# A New Method for Evaluating the Homogeneity within and between Weave Repeats in Plain Fabric Structures Using Computer Image Analysis

**DOI:** 10.3390/ma17133229

**Published:** 2024-07-01

**Authors:** Magdalena Owczarek

**Affiliations:** Faculty of Material Technologies and Textile Design, Institute of Architecture of Textiles, Lodz University of Technology, 90-924 Lodz, Poland; magdalena.owczarek@p.lodz.pl

**Keywords:** homogeneity, inter-thread pores, porosity, weave repeats, intra-repeat and inter-repeat, woven fabric structure, computer image analysis, barrier, filtration, composite

## Abstract

This article introduces a novel, rapid, and non-destructive method for assessing homogeneity within and between weave repeats in fabric structures, termed intra-repeat (IAR) and inter-repeat (IER) evaluation. The method focuses on structural parameters, including inter-thread pores (ITPs) and warp and weft pitches, using computer image analysis. Each parameter is assigned to a module in the repeat weave pattern, facilitating the sorting of modules in the IAR and IER fabric structure arrangement. The method was verified using artificial images and 30 real plain fabrics with varying degrees of warp grouping, employing the author’s proprietary software, MagFABRIC version 2.1The general measurable coefficients of intra- and inter-homogeneity were defined and related to the airflow measurements of these fabrics. Multiple regression models of airflow revealed strong dependencies, particularly for F = 10, with the size, shape, and position of ITPs and warp and weft pitches showing significant correlation. These findings underscore the importance of the new homogeneity parameters in textile structure analysis, including both IAR and IER woven fabric structure homogeneity parameters. The research aims to model specialized fabrics (e.g., barrier, filtration, composite fabrics) to address local changes in fabric structure affecting properties such as filtration efficiency, air permeability, and mechanical properties, especially in applications like composites or medical implants.

## 1. Introduction

The development of industrial civilisation puts higher and higher demands on textile products, especially those that are to be effective barriers to the flow, passage, or penetration of liquids; thermal, optical, or electromagnetic radiation; or elements of micro and macro size.

Specialised barriers and protective clothing, but also more and more often, everyday products, e.g., clothing protecting against ultraviolet (UV) radiation, are increasingly important areas of scientific activity. In the face of 21st century diseases, e.g., COVID-19, HIV, HBV, HCV, and others, the type of structure of a textile product and its uniformity have a decisive impact on the safety of society and especially emergency medical personnel who require effective barriers in their specialized clothing that prioritize safety and protection [1,2].

Another category of products includes those in which a homogeneous textile structure is required to effectively transfer internal stresses with uniformity and withstand dynamic air impacts. Examples of such products include tents, sails, and bullet-proof vests. Homogeneous textile structures are essential for ensuring the airtightness of compressed air in pneumatic cushions or the distribution of tension at spherical deformation in compositions [3,4].

In the existing literature, the homogeneity of woven fabric structures has not been consistently demonstrated in studies. It has only been inferred through the variability of average and global parameters such as warp and weft density, surface grey level, pore size, and air permeability. However, this approach does not provide adequate insight into the structure of specialized barrier fabrics. The research cited below confirms the need to conduct a precise analysis of the uniformity of the fabric structure.

Sakaguchi et al. [5] evaluated the irregularity of fabric surfaces by computing the power spectral peak width of the intensity data. Using computer image analysis, they also calculated the coefficient of variation and power spectra of yarn intervals as indices of irregularity in the yarn arrangement. However, these values were globally estimated and not separately replicated.

Kang et al. [6] introduced an automatic analysis of fabric structure utilizing computer image analysis. The authors proposed an objective assessment of various fabric structure parameters, such as the fineness and crimp of threads, cover factor, thickness, fabric areal density, and detection of fabric errors. However, they only suggested evaluating the uniformity of yarn spacing and the orthogonality of the yarn intersecting angle.

Jiraskova et al. [7] took a different approach to the issue by measuring the unevenness of the surface of woven fabric using the coefficient of variation of the grey level of the fabric image. However, this method focused solely on the surface effect of non-uniformity resulting from the non-uniformity of the yarn, rather than considering the structural analysis of the fabrics. They found that the area variation curve is a more suitable tool for identifying non-periodical irregularities. This method appears to be useful for quickly detecting errors on the fabric surface.

The inhomogeneity of a fabric structure plays a critical role in determining the uniformity of air permeability. For instance, research conducted by Havlova [8] observed a close relationship between fabric structure and air permeability. Even a minor alteration in the fabric structure at a specific location can lead to a corresponding change in permeability at that point. Higher air permeability values were associated with irregular warp pitch, corresponding to pore size variations identified through computer image analysis. Further detailed analysis of fabric structure revealed that when the fabric structure lacks regularity, using the characteristic dimension of an “average pore” may not be sufficient to predict air permeability accurately. According to the author, the crucial factor is not the average pore size but the actual size distribution of individual pores. This underscores the importance of investigating IAR and IER weave pattern fabric structure homogeneity, as it directly impacts air permeability characteristics. Ragab et al. [9] discuss determining pore size, porosity, and pore size distribution in plain weave fabric using image analysis to develop a theoretical model for predicting porosity from geometric parameters. Comparing digital surface porosity measurements with theoretical calculations reveals slight differences. Additionally, comparing pore distribution peaks with air permeability data indicates a correlation between pore spaces and air permeability.

The subsequent articles address recent research focusing on significant issues such as UV protection and filtration, particularly in light of concerns like COVID-19. These articles explore diverse methods for identifying porosity and cover fractor within fabric structures. However, despite these efforts, the analyses underscore the absence of rigorous and comprehensive identification and analysis of fabric structure parameters’ location within and outside the reported area, as well as their variability.

In a study by Kostajnšek et al. [10], porosity and cover factor were investigated concerning UV protection in woven fabric. The article utilized image analysis to illustrate the size distribution of weave pores and their contribution to inter-yarn and inter-fiber pores. The authors suggest that by knowing the yarn densities in fabrics, it is feasible to determine the number of inter-yarn pores, with the remaining pores being identified optically as inter-fiber pores. In an article by Douguet et al. [11], the relationship between air permeability and filtration efficiency was investigated as a function of the average pore area without using image analysis. The study involved 22 plain fabrics, either as a single layer or stacked, intended for COVID-19 pandemic masks with particles of 3 µm diameter. The researchers defined a model for air permeability, incorporating yarn count, calculated average inter-yarn pore area, and derived intra-yarn porosity from equations. However, it is important to note that this model represents a theoretical surface quantity and may not perfectly reflect real inter-yarn porosity, especially in the absence of hairiness. Nevertheless, the proposed model offers a practical approach for designing based on easily measurable data during manufacturing processes.

In an article by Zupin et al. [12], a significant correlation between measured and calculated porosity parameters of woven fabrics using image analysis is presented. The study conducted multifactor ANOVA statistical analysis, revealing that fabric density and weave pattern substantially impact porosity. Additionally, illumination was found to play a crucial role, whereas the threshold in the algorithm had a minor influence. Fabric images were captured using a stereomicroscope, providing a view of 30 × 16 threads in a single image, which equates to 480 inter-thread pores. The authors suggest that this method can compute the number of pores more efficiently than traditional methods, making it applicable across various industries such as clothing, medical, and technical textiles. However, Rolich et al. [13] feature research on developing thresholding algorithms used to calculate the cover factor and porosity through digital image analysis. Several algorithms are showcased to highlight the importance of this aspect of image analysis. Additionally, computational models based on machine learning were developed to efficiently predict the cover factor from fabric parameters.

In 1999, a series of research endeavors commenced, focusing on modeling channels between threads, which announced a distinct approach to comprehending fabric structure. This departure from traditional approaches highlighted the complex nature of fabric composition, extending beyond average parameters. Szosland [14] underscored the diversity and significance of channels within fabric structures, as well as the types of modulus present in fabric repeats. The author delineated four structural modules and various types of space applicable to all weaves, as shown in Figure 1, each characterized by unique geometric properties. A fabric incorporating these modules at specific locations may exhibit distinct barrier properties. It was revealed that the shape of channels in three-dimensional space, in conjunction with the fabric’s structure, raw material composition, and chemical processing phases, plays a pivotal role in shaping the barrier properties of the final product. The modeling of interstrand spaces highlighted the imperative for further research in this domain. Such investigations will facilitate the development of effective and efficient filtration and barrier structures.

Polipowski et al. [15] investigated the thread channels in woven fabric structures, building upon the modules identified by Szosland [14]. Utilizing 3D computer image analysis, the study focused on various parameters of average channels within these structures. Key parameters included channel height, spacing surface area, the angle of channel deviation from the vertical position, and surface area factor.

The preceding research underscores the imperative to investigate fabric structure homogeneity meticulously and comprehensively. Notably, no prior studies have addressed the assessment of individual parameters within IAR and IER fabric structures. To address this gap, a rapid, precise, objective, and non-destructive tool, namely the MagFABRIC program, was developed for fabric structure parameter analysis. Leveraging computer image analysis of fabric images, this tool facilitates thorough examination of fabric structure characteristics.

Preliminary research in this domain is outlined in the article by Owczarek et al. [16], where an initial examination of structure homogeneity parameters was conducted on jean-type fabrics. However, this assessment was conducted in broad terms, without distinguishing between IAR and IER homogeneity. The evaluation was centred on the uniformity of weft cover on the fabric’s left side and the consistency of weft and warp yarn diameters. The study encompassed five fabrics with structural irregularities and one reference fabric. Findings revealed that unevenness in fabrics with disturbances stemmed from irregularities in the threads, occurring across both short (2, 10 [mm]) and long sections (50, 100 [mm]). These irregularities led to noticeable disruptions in fabric structure. This approach aligns with contemporary expectations for the speedy and accurate non-destructive and non-subjective assessment of a finished product’s quality.

The research aims to objectively and quantifiably establish new parameters for describing and evaluating homogeneity within and between weave repeats in fabric structures. An innovative methodology was devised for identifying and characterizing individual construction parameters assigned to each module of the weave repeat, employing computer image analysis.

## 2. Materials and Methods

### 2.1. Methods

This research focuses on evaluating the homogeneity within and between repeats of the woven fabric structure, specifically examining the consistency of structural parameters within and between repeats. These new parameters are understood as the repeatability of the smallest elementary unit—the weave repeat (intra-repeat (IAR)) and the repeatability of the collection of these elementary units—a collection of repeats (inter-repeat (IER)), as shown in Figure 2.

The plain weave repeat, as shown in Figure 3a, traditionally consists of two warp overlaps, two weft overlaps, and one full, four half, and four quarter ITPs. Evaluating the homogeneity of the plain weave fabric structure within and between repeats required defining a repeat that encompasses all ITPs, as shown in Figure 3b. This entailed identifying four weave structural modules of the SMS1 type {1d, 2, 3d, 4}, each oriented oppositely, based on the research by Szosland [14] as shown in (Figure 3c).

The homogeneity of the IAR is defined as the repeatability of every structural parameter (including the size, shape, and location of the ITPs, as well as the value and position of the warp and weft pitches) within each fabric weave repeat, assigned to every modulus SMS1 {1d, 2, 3d, 4} and sorted as shown in Figure 4. 

The homogeneity of the IER is defined as the repeatability of the same structural parameters across the collection of repeats, assigned and sorted precisely according to every number of the modulus SMS1 {1d, 2, 3d, 4} from every image of the fabric, as shown in Figure 5.

Warp/weft pitch refers to the distance between individual warp/weft yarns in a woven fabric, indicating the spacing or arrangement of the warp/weft yarns. It directly relates to the spacing of the warp yarns along the length of the fabric.

The methodology for evaluating fabric structure homogeneity relies on several key parameters assigned to individual repeats:The size, shape, and location of ITPs;The value and position of thread pitches.

Structural analysis of ITPs and thread pitches in woven fabrics was conducted using the author’s MagFABRIC software, which is based on computer image analysis of the author’s ITP morphometric analysis by Owczarek [17]. The main parameters and essential concepts crucial for the homogeneity method are depicted in Figure 6 and elaborated below.

Equation of parameters important for the uniformity of the fabric structure.

The ITP area (A) is described as follows:(1)$$A=\sum_{i=1}^{n}\sum_{j=1}^{m}p(i,j);[\mathrm{pix}]$$
where p(i, j)—pixel of the ITP area; i, j—coordinates of the Cartesian space of the image; n, m—resolution of the image.

The ITP shape (S) is determined as follows:(2)S=0.45×Feret+0.45×AspecR+0.1×FormF;[-]
(3)S1=(Feret+AspecR+FormF)/3;[-]
where equations S and S1 with different weights of constant coefficients (0.45, 0.1, or 0.33) were verified by multiple regression in Model 1. The coefficients were chosen based on theoretical assumptions regarding the weights of individual coefficients for the overall assessment of ITP shape.

The Feret degree of elongation: (0 ≤ Feret < 1) → vertical elongation, (Feret ≈ 1) → square, (1 < Feret < ∞)→ horizontal can be expressed as follows:(4)$$Feret=\frac{S}{W};[\text{-}]$$
where S is the ITP area width and W is the ITP area height.

The AspectR degree of oval shape: (0 ≤ AspectR < 1) → elliptically flattened (AspectR ≈ 1) → oval can be expressed as
(5)$$AspecR=\frac{D_{MIN}}{D_{Max}};[\text{-}]$$
where D_MIN_ is the ITP minimum area diameter and D_MAX_ is the ITP maximum area diameter.

The FormF is the degree of development of the edges: (0 ≤ FormF < 1) → (1 = not corrugated; 0 = very corrugated), and can be expressed as
(6)$$FormF=\frac{4\Pi \times A\text{\_}ITP}{L^{2}};[\text{-}]$$
where L is the ITP perimeter.

The Warp pitch (P_wa_) is the distance between the axes of adjacent warp threads.

The Weft pitch (P_we_) is the distance between axes of adjacent weft threads:(7)$$P_{wa}=p_{\max}\left(x+1\right)-p_{max}\left(x\right);[\mathrm{pix}]$$
(8)$$P_{we}=p_{\max}(y+1)-p_{max}(y);[pix]$$
(9)$$p\left(x\right)=\frac{1}{1024}\sum_{i=0}^{1024}p\left(x,y_{i}\right);\left[pix\right]$$
(10)$$p(y)=\frac{1}{1024}\sum_{i=0}^{1024}p\left(x_{i},y\right);\left[pix\right]$$
where p_max_(x) and p_max_(y) are the local maxima of p(x) and p(y) profiles—functions with a variable period depending on the pitch of the threads.

The ITP Position (D_IDE_) concerning the averaged bimodal grid: (0 ≤ D < 1) → (1 = close distance, 0 = far distance) is as follows:(11)$$D_{IDE}=\frac{\bar{P}r-C}{\bar{P}r};[\mathrm{pix}]$$
(12)$${\bar{P}}_{r}=\sqrt{{\bar{P_{wa}}}^{2}+{\bar{P_{we}}}^{2}}$$
where D_IDE_ is the distance of the C centre of gravity of the ITP from the nearest intersection of the averaged grid of pitches; ${\bar{P}}_{r}$ is the mean diagonal of the pitch’s rectangle from P_wa_ and P_we_.

The RID is the difference between the ITP area (A_ITP_) and the ideal area of the averaged grid $\left({\bar{P}}_{IDE}\right)$; (0 ≤ RID < 1) → (1 = very good fit, 0 = large offset from the grid) and can be expressed as follows:(13)$$RID=\frac{A_{ITP}}{{\bar{P}}_{IDE}};[pix]$$

All parameters must be assigned to the structural modules; for example, each ITP has its own warp and weft pitch, as well as size, shape, and location. This allows for the uniform sorting of the ITPs within the fabric repeats. To measure IAR inhomogeneity (${\bar{V}}_{IAR}$), the structural modules SMS1{1d, 2, 3d, 4} are sorted repeats (Figure 5). Conversely, for the measurement of IER inhomogeneity (${\bar{V}}_{IER}$) of the structural modules SMS1{1d, 2, 3d, 4}, the data are sorted according to belonging to individual structural modules (Figure 6).

#### 2.1.1. The General Coefficient of IAR Inhomogeneity (${\bar{V}}_{IAR}$)

The methodology for calculating the coefficient of IAR inhomogeneity (${\bar{V}}_{IAR}$) of the fabric structure involved measuring the average coefficient of variation of a given parameter across modules within the repeats, denoted as {V_IAR1_(M),…,V_IARn_(M)}:(14)$${\bar{V}}_{IAR}=\frac{1}{n}\sum_{i=1}^{n}V_{IARi}(M);[\%]$$

The general coefficient of intra-repeat inhomogeneity (${\bar{V}}_{IAR}$) determines the level of IAR inhomogeneity based on the following parameters:The coefficients of the ITP variation:1.1V_IAR__A—the IAR inhomogeneity of the ITP area (A);1.2V_IAR__S—the IAR inhomogeneity of the ITP shape (S);V_IAR__Feret—the IAR inhomogeneity of the ITP elongation (Feret);V_IAR__AspR—the IAR inhomogeneity of the ITP oval shape (AspectR);V_IAR__FormF—the IAR inhomogeneity of the edge development of the ITP (FormF).The coefficients of the thread pitch variation:2.1V_IAR__P_wa_—the IAR inhomogeneity of the warp thread pitches (P_wa_);2.2V_IAR__P_we_—the IAR inhomogeneity of the weft thread pitches (P_we_).The coefficients of the average grid variation:3.1V_IAR__D_ITP_—the IAR inhomogeneity of the position (D_ITP_);3.2V_IAR__RID—the IAR inhomogeneity of the area difference (RID).

The general coefficient of IAR inhomogeneity (${\bar{V}}_{IAR}$) was adopted from three propositions:(15)$${\bar{V}}_{IAR1}=0.2\times V_{\mathrm{IAR}}\text{\_}A+0.2\times V_{\mathrm{IAR}}\text{\_}S+0.2\times V_{\mathrm{IAR}}\text{\_}P_{\mathrm{wa}}+0.2\times V_{\mathrm{IAR}}\text{\_}P_{\mathrm{we}}+0.1\times V_{\mathrm{IAR}}\text{\_}D_{\mathrm{ITP}}+0.1\times V_{\mathrm{IAR}}\text{\_}\mathrm{RID}\;[\%]$$
where V_IAR__S according to Equation (2);
(16)$${\bar{V}}_{IAR2}=(V_{\mathrm{IAR}}\text{\_}A+V_{\mathrm{IAR}}\text{\_}S+V_{\mathrm{IAR}}\text{\_}P_{\mathrm{wa}}+V_{\mathrm{IAR}}\text{\_}P_{\mathrm{we}}+V_{\mathrm{IAR}}\text{\_}D_{\mathrm{ITP}}+V_{\mathrm{IAR}}\text{\_}\mathrm{RID})/6\;[\%]$$
where V_IAR__S according to Equation (2); and
(17)$${\bar{V}}_{IAR3}=(V_{\mathrm{IAR}}\text{\_}A+V_{\mathrm{IAR}}\text{\_}S1+V_{\mathrm{IAR}}\text{\_}P_{\mathrm{wa}}+V_{\mathrm{IAR}}\text{\_}P_{\mathrm{we}}+V_{\mathrm{IAR}}\text{\_}D_{\mathrm{ITP}}+V_{\mathrm{IAR}}\text{\_}\mathrm{RID})/6\;[\%]$$
where V_IAR__S1 according to Equation (3).

The values of the coefficients 0.2 and 0.1 were chosen based on theoretical assumptions regarding the weights of individual coefficients for the overall assessment of IAR inhomogeneity $({\bar{V}}_{IAR}$). The IAR inhomogeneity $\left({\bar{V}}_{IAR1},{\bar{V}}_{IAR2},{\bar{V}}_{IAR3}\right)$ was verified using the multiple regression Model 2 in which, for this model, the equation ${\bar{V}}_{IAR1}$ was the best correlation.

#### 2.1.2. The General Coefficient of Inter-Repeat Inhomogeneity (${\bar{V}}_{IER}$)

The methodology for calculating the IER inhomogeneity (${\bar{V}}_{IER}$) of the fabric structure was determined using two methods: “elements” and “averages”. In the first method, individual modules from the repeats were sorted, and from them, the average variability of a given parameter was determined as {V_IER_(1d),…, V_IER_ (4)}:(18)$${\bar{V}}_{IER}(E)=\frac{1}{4}\sum_{M=1}^{4}(V_{\mathrm{IER}}(1D),\ldots ,V_{IET}(4))\;[\%]$$

On the other hand, in the method of averages, the averages from the repeats were determined, and their average volatility was calculated based on these averages:(19)$${\bar{V}}_{IER}(A)=\frac{\delta_{IER}}{{\bar{X}}_{IER}}\times 100\;[\%]$$

The general coefficient of inter-repeat inhomogeneity (${\bar{V}}_{IER}$) determines the level of IER inhomogeneity based on the following parameters:The coefficients of the ITP variation:1.1V_IER__A—the IER inhomogeneity of the ITP area (A);1.2V_IER__S—the IER inhomogeneity of the ITP shape (S);V_IER__Feret—the IER inhomogeneity of the ITP elongation (Feret);V_IER__AspR—the IER inhomogeneity of the ITP oval shape (AspectR);V_IER__FormF—the IER inhomogeneity of the edge development of the ITP (FormF).The coefficients of the thread pitch variation:2.1V_IER__P_wa_—the IER inhomogeneity of the warp thread pitches (P_wa_);2.2V_IER__P_we_—the IER inhomogeneity of the weft thread pitches (P_we_).The coefficients of the average grid variation:3.1V_IER__D_ITP_—the IER inhomogeneity of the position (D_ITP_);3.2V_IER__RID—the IER inhomogeneity of the area difference (RID).

The general coefficient of IER inhomogeneity (${\bar{V}}_{IER}$) was adopted from three propositions:(20)$${\bar{V}}_{IER1}=0.2\times V_{\mathrm{IER}}\text{\_}A+0.2\times V_{\mathrm{IER}}\text{\_}S+0.2\times V_{\mathrm{IER}}\text{\_}P_{\mathrm{wa}}+0.2\times V_{\mathrm{IER}}\text{\_}P_{\mathrm{we}}+0.1\times V_{\mathrm{IER}}\text{\_}D_{\mathrm{ITP}}+0.1\times V_{\mathrm{IER}}\text{\_}\mathrm{RID}\;[\%]$$
where V_IER__S according to Equation (2);
(21)$${\bar{V}}_{IER2}=(V_{\mathrm{IER}}\text{\_}A+V_{\mathrm{IER}}\text{\_}S+V_{\mathrm{IER}}\text{\_}P_{\mathrm{wa}}+V_{\mathrm{IER}}\text{\_}P_{\mathrm{we}}+V_{\mathrm{IER}}\text{\_}D_{\mathrm{ITP}}+V_{\mathrm{IER}}\text{\_}\mathrm{RID})/6\;[\%]$$
where V_IER__S according to Equation (2); and
(22)$${\bar{V}}_{IER3}=(V_{\mathrm{IER}}\text{\_}A+V_{\mathrm{IER}}\text{\_}S1+V_{\mathrm{IER}}\text{\_}P_{\mathrm{wa}}+V_{\mathrm{IER}}\text{\_}P_{\mathrm{we}}+V_{\mathrm{IER}}\text{\_}D_{\mathrm{ITP}}+V_{\mathrm{IER}}\text{\_}\mathrm{RID})/6\;[\%]$$
where V_IER__S1 according to Equation (3).

The values of the coefficients 0.2 and 0.1 were selected based on theoretical assumptions regarding the weights of individual coefficients for the overall assessment of of IER inhomogeneity $({\bar{V}}_{IER}$). This coefficient was validated for both the “elements” method and the “averages” method using theoretical model fabric images. The IER inhomogeneity $\left({\bar{V}}_{IER1},{\bar{V}}_{IER2},{\bar{V}}_{IER3}\right)$ was verified using the multiple regression Model 2, where the equation (${\bar{V}}_{IER2})$ exhibited the best correlation.

### 2.2. Material

#### 2.2.1. The Images of Theoretical Model Fabrics for Verification of the New Method

For the initial verification of the new method, theoretical fabric model images were generated using graphical software on a computer. These images featured varied geometric elements designed to assess parameters of inhomogeneity related to size, shape, and position of ITPs. Individual repeats were intentionally inverted to evaluate the sensitivity of the methodology parameters. The models depicted in Figure 7 were created assuming that the geometric shapes of internal objects mimicked real ITPs, featuring diverse locations, sizes, and shapes to evaluate the sensitivity of the developed methodology to these variations. For instance, different aspect ratios corresponded to specific shape characteristics such as ellipticity, elongation, or the degree of coastline development. Consequently, the final homogeneity model incorporated three factors sensitive to such variability. The measured parameters of the theoretical fabric models are presented in Table 1.

#### 2.2.2. The Images of Plain Weave Fabrics for Verification of the New Method

The 30 plain weave fabrics were produced on a laboratory Saurer 100 W Shuttle Loom by Saurer Group by adjusting several loom setting parameters including pre-tension of the warp (5.93–31.91 cN/thread), the backrest roller position (+4 to −4 cm from the center position), the moment of closing the shed adjusted from open or closed to crossed shed, and the lease rod position set between 43 and 73 cm from the geometric center of the harness. The warp density was maintained at 230 ends/dm, while the weft density was 130 picks/dm. These fabrics were manufactured using combed cotton and two-ply ring yarn with 646 twist/m and linear mass of 20 × 2 Tex. More parameters of yarns are shown in Table 2.

The obtained 30 fabrics have the same weave and surface mass. They differ only in the size of the grouping of warp threads. Variable parameters of the weaving process in each fabric allowed for different structures of the plain weave fabric to be obtained. The grouping of warps is the result of the conditions prevailing in the area of fabric formation on the loom, particularly the ability of the structure to self-regulate. Different degrees of grouping are shown in the images of the structures of plain cotton fabrics in Figure 8. The measured parameters of the plain fabric are presented in Table 3.

The varying degrees of grouping of the warp threads determine the different structures of these fabrics and, consequently, their parameters, including the size, shape, and position of ITPs, as well as the size and position of the pitches of the warp and weft threads. This diversity is not clearly reflected by the basic structural parameters, and unfortunately, it results in variable properties, affecting both filtration and strength properties.

The fabrics obtained from the experiment were subjected to conditioning and acquisition under constant, normal conditions, as shown in Figure 8. For each fabric, 30 images were captured along a diagonal line across the entire width of the fabric 1.5 m long. The author’s program MagFABRIC allows one to stitch several images together to enlarge the fabric surface while maintaining the highest possible accuracy at a magnification of 3 × 3 mm^2^. Fabric image acquisition was performed using a stereoscopic microscope, MST Zoom 1302 CB, and a CCD-4012 camera. The acquisition parameters included a 1.25 magnification zoom, capturing a 3 × 3 mm^2^ area of the woven fabric in the image, representing approximately 8 × 5 threads with about 28–36 pores. The spatial resolution was set at 1024 × 1024, with grey levels ranging from 0 to 255, and illumination was provided by passing light through an optical fiber ring, specifically the Olympus Highlight 3100. The accuracy of individual pattern dimensions was verified using a Mitutoyo projector at a magnification of 10x, with a calculated accuracy of 0.001 mm. Subsequently, the patterns were utilized for acquisition and analysis in the MagFABRIC version 2.1 software.

The research commenced with the assessment of radiometric and geometric distortions, as well as irregularities in image brightness. Subsequently, an algorithm comprising various procedures was implemented, encompassing image pre-processing, segmentation and recognition, classification, and interpretation based on cluster analysis. The image processing algorithm includes acquisition in the same light parameters and after the conditioning process in normal conditions (humidity 60%, temperature 20 °C), low-pass filtering, histogram equalization to the full range h [0–255], nonlinear filtration by the square filter, image negative, thresholding operation with the auto threshold set by copyright procedure, closing and opening operation, and ITP structural analysis, as shown in Figure 9. Fabric image pre-processing plays a pivotal role in optimizing the algorithm for detecting individual ITPs. Here are some key aspects highlighting its impact and the great importance described in the earlier article by Owczarek [18]. Noise reduction: Pre-processing techniques such as low-pass filtering and nonlinear filtration help reduce noise and artifacts present in fabric images. This is crucial for enhancing the clarity of the image and improving the accuracy of ITP detection. Enhancement of contrast and brightness: Histogram equalization enhances the contrast of the image by stretching the intensity values across the entire dynamic range. This ensures that subtle features, including ITPs, are more distinguishable from the background. Thresholding for segmentation: Automatic thresholding based on the characteristics of the image is essential for accurate segmentation of ITPs. By determining an optimal threshold value, pre-processing enables the algorithm to effectively separate the foreground (ITPs) from the background. The development of an auto threshold set was of particular significance, utilizing two region-splitting methods based on the distributions of background p(x,y) and object f(x,y) in each image. This algorithm, employing segmentation based on Gauss and Poisson methods, facilitated the determination of an optimal threshold value for segmentation. This approach was instrumental in cross-validating the sought segmentation threshold. Morphological operations: Closing and opening operations help refine the segmented regions and eliminate small imperfections or gaps. This ensures that detected ITPs are well-defined and continuous structures, enhancing the accuracy of subsequent analysis. Overall image quality: Pre-processing techniques collectively contribute to improving the overall quality of the fabric image. This ensures that the algorithm operates on images with consistent brightness, contrast, and clarity, leading to more reliable and robust detection of ITPs. In summary, fabric image pre-processing plays a critical role in optimizing the algorithm for individual inter-thread pore detection by enhancing image quality, reducing noise, improving contrast, and enabling accurate segmentation of ITPs.

Cluster analysis has been used to identify and segment ITPs in fabric images. Its operation algorithm is multi-stage. A clustering algorithm is applied to the data points to divide them into groups or clusters based on similarity. The original program initially used the region-growing segregation method, where pixels were tested according to their degree of similarity. The uniformity criteria were the segmentation threshold and radius. However, the method did not produce the expected results. The disadvantages of this method have been confirmed. A big problem was determining the segmentation radius that would separate object areas. There is, for example, thinning and thickening of threads in the fabric, which brings the gaps closer or further apart. In such a case, an incorrectly selected radius causes the gaps between the threads to be glued together or fragmented into smaller objects. An original, multi-stage grouping was introduced, taking into account the textile structure’s features and location. After applying the clustering algorithm, indexing occurs; i.e., each data point is assigned to a specific cluster. Pixels belonging to the same cluster are grouped, allowing ITPs to be isolated and separated from the background. In the context of ITP identification, clusters may represent areas of the image corresponding to individual ITPs or groups of ITPs in a weave repeat. Finally, the segmented ITPs can be visualized and analysed to extract relevant information on their size, shape, distribution, and other morphometric features and parameters. This information can be used for further analysis to quantify the quality of the fabric structure. Cluster analysis with the author’s modernization provides an optimal approach for identifying and segmenting ITPs in fabric images, enabling automatic or semi-automatic analysis of fabric structure with high efficiency and accuracy.

## 3. Results and Discussion

### 3.1. Verification of the New Method Using the Images of Theoretical Model Fabrics

The verification of IAR and IER inhomogeneity parameters showed differences in the images. Based on the graphs (Figure 10 and Figure 11), we can see that the best homogeneity within and between the repeats was shown by Model 14. The highest unevenness for this image, only 6.5%, was the parameter characterising the weft thread pitches (V_IER__P_we_), as shown in Figure 10 and Figure 11, both in the intra- and inter-repeat.

Models 17, 19, 111, and, to a lesser extent, 18 are the most inhomogeneous images in terms of IER assessment, as shown in high shape parameters in Figure 10a. These are cases of images where the repeat was intentionally reversed to cause a large disturbance. Model 111 significantly stands out in the analysis results with its shape parameters. As the only image, it has the most distorted forms of structural elements, and this was well captured by the parameter (V_IER__FormF = 27%), which characterises the degree of edge development. Also, we have the highest inhomogeneity parameter for this model between the elongation of the weave element (V_IER__Feret = 42%). The remaining images have a similar level of successive indices of IER homogeneity. These are image models where no-repeat rotation has been performed. The (V_IER__P_we_) thread pitch uniformity parameter for these images stands out from the rest of the parameters. This proves the differences in the vertical position of the entire repeat. Model 112 is noteworthy because it has the highest (V_IER__P_we_) index of 13%, which is caused by the presence of various shapes of elements with different locations of centres of gravity in the image.

The verification of IER inhomogeneity methods revealed differences in images, as expected. Information on IER inhomogeneity in the case of the “elements” method is more accurate, as shown in Figure 10a. For instance, the “average” method failed to detect the unevenness resulting from the reversal of the repeat in Models 18 and 111 and differences in the size and shape of the ITPs, as shown in Figure 10b. This discrepancy was effectively demonstrated by the “elements” method. In Models 17 and 19, the “average” method indicated only a sensitivity of 4.5% inhomogeneity of the (V_IER__D_ITP_) parameter, suggesting that the centers of the ITP elements did not align on one line of the two-modal grid, possibly indicating variations in the shapes and sizes of these elements. In contrast, the “elements” method more accurately described the differences between these elements across repeats. It was found that the greatest inhomogeneity in Models 17, 19, and 111 occurred in the ITP area (V_IER__A) and its shape (V_IER__S, V_IER__Feret, V_IER__AspR, and V_IER__FormF), averaging 35%. Additionally, parameters related to the ITP area and shape (V_IER__RID) exhibited the most significant differences, indicating variations between the ITP area and shape across repeats. These findings are consistent with the conceptualization of the images created in Models 17, 19, and 111.

The analysis of IAR inhomogeneity delineated differences among artificial images, particularly regarding the size, shape, and position of ITPs. The longest peaks in the chart represent the IAR inhomogeneity of the ITP area (V_IAR__A) for Models 11 and 13, reaching around a 60% coefficient of variation, as depicted in Figure 11. Exactly these images exhibit significant variation in the size of the ITP area within the repeat. The next two high peaks in the graph are due to the elongation coefficient (V_IAR__Feret), which is very sensitive to the rotation of the bodies, which was made in Models 16 and 110. The ovality coefficient (V_IAR__AspR), in these images did not change because the bodies were only rotated. The remaining image Models: 15, 17, 19, 111, and 112 demonstrated inhomogeneity in both area size and shape within the range of (30–40%), with Model 18 (20%) exhibiting slightly lesser deviations. Similar to the assessment of IER inhomogeneity, Model 14 showed the best assessment in the case of IAR homogeneity. This is confirmed by the graph in Figure 10a and Figure 11).

The verification of the general coefficient of IAR and IER inhomogeneity (${\bar{V}}_{IAR},{\bar{V}}_{IER}$) determined by formulas (15 and 20, respectively), effectively reflects the variability of artificial images and confirms the variability of individual parameters such as the size, shape, and location of the ITPs, as described above. Consistently with the graphs in Figure 10, Figure 11 and Figure 12, the image with the best homogeneity (${\bar{V}}_{IAR}=1.75\%$, ${\bar{V}}_{IER}=1.69\%$) is Model 14, exhibiting a uniform field, shape, and position of the ITPs in the form of a square without rotation or shape change. Conversely, Model 17 exhibits the highest level of inhomogeneity within (${\bar{V}}_{IAR}=19.17\%$) and between repeats (${\bar{V}}_{IER}=19.59\%$), with its IER variability demonstrating the sensitivity of the parameter between repeats determined by the “element” method. Lastly, Model 112 is distinguished in the analysis, showcasing high intra-report inhomogeneity (${\bar{V}}_{IAR}=22.99\%$) but low inter-report inhomogeneity (${\bar{V}}_{IER}=3.63\%$) using the developed general parameters. This result confirms the degree of variation in ITP shape within the report of this image and the lack of variation in repeats between reports. Models 11, 13, 15, 16, and 110 had high IAR inhomogeneity but obtained results of good IER homogeneity. Furthermore, the IAR inhomogeneity was correctly estimated on the artificial images. In the case of IER inhomogeneity, the accuracy of two methods was determined: “elements” and “averages”. The accuracy of the “average” method was found to be very low, so the “element” method, which accurately responds to various variations, was selected for further research.

### 3.2. Verification of the New Method Using the Images of Plain Weave Fabrics

Based on the computer analysis of the fabric images, the results of the fabric structure parameters were obtained and analysed using new methods for assessing intra-repeat and IER homogeneity. Changes in input parameters on the weaving loom resulted in different structure homogeneity for fabrics of the same type of weaving. Changes in the loom setting parameters during fabric manufacturing generated a different degree of self-regulation of the fabric structure, and thus a different degree of its homogeneity. Three characteristic groups of structures can be distinguished based on the degree of warp grouping, as shown in Figure 13 and Figure 14.

In the first group of fabrics, the fabric structure exhibits the most even IAR and IER structure. This means that the size, shape, and position of the ITPs, as well as the value and position of the warp and weft pitches, are regular within and between every repeat. Additionally, the warp threads in this fabric are not grouped, indicating a possibility for autoregulation during the weaving process manufacturing. Fabric P_5 serves as an example of this group, demonstrating the lowest inhomogeneity coefficients, with (${\bar{V}}_{IAR}=26.33\%$) and (${\bar{V}}_{IER}=24.91\%$), as indicated by the green result in Figure 13.

The second group comprises fabrics with a characteristic visible trace of reed in the form of warp grouping. Unlike the first group, the average grid is much more bimodal and these fabrics show greater differences in bimodal ITP areas regarding size, shape, and position. Additionally, there is a more noticeable difference in the value of warp pitches compared to fabrics in the first group, such as P_5. In this case, the warp threads are grouped, indicating a lack of possibility for autoregulation during the weaving process manufacturing. Despite exhibiting very good IER homogeneity, these fabrics have much worse IAR homogeneity. This suggests an inability to self-regulate the structure, with direct and strong stabilisation occurring in the area of weaving. Fabric P_20 serves as an example, with the lowest IER inhomogeneity coefficient (${\bar{V}}_{IER}=24.52\%$) and a higher IAR inhomogeneity coefficient (${\bar{V}}_{IAR}=34.99\%$), as shown by the yellow result in Figure 13.

The third group encompasses fabrics with fabric structure inhomogeneity in both intra- and inter-repeats. These fabrics exhibit highly visible warp grouping and the greatest differences in bimodal size, shape, and position of the ITP areas, particularly in the value and position of the warp pitches. In this case, the warp threads are heavily grouped, indicating a lack of possibility for autoregulation during the weaving process manufacturing. These fabrics demonstrate the worst intra-repeat and IER homogeneity, suggesting an inability to self-regulate the structure. However, there are attempts at self-regulation, albeit with strong stabilization occurring in the area of weaving. These attempts are evident through the observed inhomogeneity. A prime example is the worst fabric, P_12, with the highest inhomogeneity coefficients (${\bar{V}}_{IAR}=40.68\%$) and (${\bar{V}}_{IAR}=35.31\%$), as depicted by the red result in Figure 13.

The obtained results of homogeneity were related to the actual measurement of airflow, which was carried out for 30 plain weave fabrics using the CEN. (2024) [19] and ASTM. (2018) [20] standards with the FX 3300 Air Permeability Tester III device from TEXTEST Instruments. The research area was determined by the 20 mm^2^ circular testing head. The results show a varied level of this parameter for this group of textiles, which exactly coincides with the results of the average inhomogeneity V of these woven structures, as indicated by the red line for AirF and the blue line for V in Figure 13. Similarly, three characteristic plain fabrics with the lowest inhomogeneity (P_5) have the lowest airflow value of AirF = 383.73 [mm/s]. Conversely, the fabric with the highest inhomogeneity (P_12) has the highest airflow of AirF = 620.91 [mm/s]. Fabric P_20, with the lowest IER homogeneity and worst IAR homogeneity, has an average air permeability of AirF = 510.73 [mm/s], as shown by the red line in Figure 13 and on the images in Figure 14.

### 3.3. Verification of the New Method Using the Pore Size Distribution

The obtained results of homogeneity were related to the measurement of pore size distribution, which was carried out for 30 plain weave fabrics using image analysis. For each fabric, every ITP from 30 images was taken, sorted, and divided into 30 divisions. Histograms of pore size distribution for each fabric were created, showing the frequency of each pore ITP range distribution. The range was calculated as a common value for all fabrics.

Figure 15 shows the pore size distribution for three characteristic plain fabrics after image analysis: P_5, P_12, and P_20. The fabric P_5, with the best IAR and IER homogeneity according to the new method, has regular, near-average-sized pores—not too big, nor too small—in the global population on the histogram of pore size distribution. Fabric P_12, with the worst homogeneity, also correlated with its pore size distribution, featuring the most pores of the smallest size and covering all size ranges. Fabric P_20, with the best IER homogeneity but the worst IAR homogeneity, also shows a full range of pore size distribution but with a predominance of smaller sizes.

The statistical description of the pore size distribution for each plain fabric does not indicate which fabric has the best pore size distribution, as shown in Table 4. The lowest standard deviation σ (A) shows that fabrics P_5, P_7, P_6, P_8, P_21, and P_17, have a regular size distribution near 2300 pixels. The lowest median values indicate that fabrics P_15 and P_2 have the lowest central values of the set, but the lowest average pore size is found only in P_18. Therefore, from these statistical descriptions, predicting which fabric has the best pore size distribution and homogeneity is difficult, especially the inter-repeat and IAR homogeneity of the pore size. The statistical description of the pore size distribution gives a general description of the overall distribution.

### 3.4. Verification of the New Method Using the Results of the Multiple Regression Analysis for the Air Flow (AirF) and ${\bar{V}}_{IAR}$ and ${\bar{V}}_{IER}$ Homogeneity Structure Parameters

The multiple regression analysis aimed to explore the relationship between air flow AirF [mm/s] as the output variable and various parameters of plain fabric structure inhomogeneity, including both intra-repeat V¯IAR and inter-repeat V¯IER (V_IAR__A, V_IAR__Feret, V_IAR__AspR, V_IAR__FormF, V_IAR__P_wa_, V_IAR__P_we_, V_IAR__D_,_ V_IAR__RID, V_IAR__S, V_IER__A, V_IER__Feret, V_IER__AspR, V_IER__FormF, V_IER__P_wa_, V_IER__P_we_, V_IER__D_,_ V_IER__RID, and V_IER__S) as input variables. The stepwise progressive multiple regression method was employed using Statistica software, with F values ranging from 1 to 10 for the verification of stronger dependencies. The input variables consisted of a set of 54 variables, including their first to third power forms and their 108 mutual interactions.AirF = 105.99 − (6.69 × V_IAR__P_wa_) − (0.0003 × V_IER__P_we_^2^ × V_IER__AspR) + (6.41 × V_IAR__A)+(0.72 × (0.45 V_IER__Feret + 0.45 V_IER__AspR + 0.1 V_IER__FormF)(Model 1)

In Model 1, an (R^2^) value of 0.86 was obtained, indicating that 86% of the total variability of the AirF variable is explained by the model at F = 1. The (R^2) value of 0.84 suggests that 84% of our equation would fit another sample from the same population. The strongest connections between individual independent variables and the dependent variable AirF are observed where the highest value of the standardized BETA coefficient occurs, and in these cases, the p value does not exceed the assumed level of 0.00 for (V_IAR__A) (BETA = 1.39), *p* = 0.0013 and (V_IER__P_we_^2^ × V_IER__AspR) (BETA = −0.43), *p* = 0.0000.AirF = 125.55 − (4.92 × V_IAR__P_wa_) − (0.0003 × V_IER__P_we_^2^ × V_IER__AspR) + (6.11 × V_IAR__A)(Model 2)

In Model 2, (R^2^) = 0.83 and adjusted (R^2) = 0.82 with F = 5. The strongest connections between individual independent variables and the dependent variable AirF are observed for (V_IER__P_we_^2^ × V_IER__AspR) (BETA = −0.41), V_IAR__A (BETA = 1.32), *p* = 0.0000.AirF = 302.21 + (11.82 × V_IAR__P_wa_)(Model 3)

In Model 3, (R^2^) = 0.71 and adjusted (R^2) = 0.70 with F = 10. The strongest connections between individual independent variables and the dependent variable AirF are observed in these cases for V_IAR__P_wa_ (BETA = 0.845), *p* = 0.0000.

From the multiple regression Model 1, the most important parameters of fabric structure homogeneity influencing the airflow parameter were identified. Among the IAR inhomogeneity parameters, the strongest correlation was observed with the ITP size (V_IAR__A) and warp thread pitches (V_IAR__P_wa_), while among the IER inhomogeneity parameters, the interaction of weft thread pitches (V_IER__P_we_) and ITP aspect ratio (V_IER__AspR) showed the strongest correlation, and also sum with weights of three shape ITP parameters: (V_IER__Feret) ITP elongation Feret, (V_IER__AspR) ITP oval shape AspectR and (V_IER__FormF) edge development of the ITP FormF, which was described as the S equation for the shape of the ITP area (2) from among the proposed S and S1, which are presented in Equations (2) and (3).

For each model, the strongest correlation of inhomogeneity parameters was checked. The value of F was increased incrementally from F = 1 through 5 up to 10. Corresponding to these changes, the value of (R^2) decreased from 0.84 to 0.82 and 0.70, indicating a slightly lower degree of fit to a similar study population. In Models 1 and 2, we obtained significance for the inhomogeneity parameters of ITP size (V_IAR__A), weft pitches (V_IER__P_we_), and ITP shape (V_IER__AspR). Model 3, at F = 10, highlighted only one strongest variable, the warp pitches (V_IAR__P_wa_), which also appeared in Models 1 and 2.

The obtained modeling validates the significance of the new parameters of textile structure homogeneity, particularly in assessing both the weft and warp pitches, as well as the size and shape of the ITP area. Additionally, the models identified the most important dependencies for the shape of the ITP area among the proposed equations. An important conclusion drawn from the models is that it indicates both IAR and IER inhomogeneity fabric structure parameters, underscoring the comprehensive nature of the analysis.

### 3.5. Verification of the New Method Using the Multiple Regression Models of the Air Flow AirF [mm/s] and the General Coefficient of IAR and IER Inhomogeneity (${\bar{V}}_{IAR}{,\bar{V}}_{IER}$)

The multiple regression analysis aimed to predict the air flow AirF [mm/s] (output variable) based on the main plain fabric structure inhomogeneity parameters within and between repeats. These parameters include ${\bar{V}}_{IAR1}\left(15\right),{\bar{V}}_{IAR2}\left(16\right),{\bar{V}}_{IAR3}\left(17\right),{\bar{V}}_{IER1}\left(19\right),{\bar{V}}_{IER2}\left(20\right),{\bar{V}}_{IER3}(21)$ (input variables). The input variables comprised 18 parameters in the form of the first to the third power, along with their 18 mutual interactions. The strongest dependencies for F = 10 are presented below.
(Model 4)$$\mathrm{AirF}=105.53+(13.34\times {\bar{V}}_{IAR1})-(3.98\times {\bar{V}}_{IER2})$$
where 

${\bar{V}}_{IAR1}$ = 0.2 × V_IAR__A + 0.2 × V_IAR__S + 0.2 × V_IAR__P_wa_ + 0.2 × V_IAR__P_we_ + 0.1 × V_IAR__D_ITP_ + 0.1 × V_IAR__RID according to Equation (15),

${\bar{V}}_{IER2}$ = (V_IER__A + V_IER__S + V_IER__P_wa_ + V_IER__P_we_ + V_IER__D_ITP_ + V_IER__RID)/6 according to Equation (20),

and V_IAR__S, V_IER__S according to Equation (2).

In Model 4, R^2^ = 0.81, and adjusted R^2 = 0.79 with F = 10. Among the independent variables, the most significant correlations with the dependent variable AirF are as follows: (${\bar{V}}_{IAR1}$) (BETA = 1.00, p = 0.0000) and (${\bar{V}}_{IER2}$) (BETA = − 0.31, p = 0.0031). The most significant correlations were obtained from the multiple regression Model 4 with the main equations of the overall IAR and IER inhomogeneity structural fabric parameters, which have a significant impact on the air flow parameter: the IAR and IER inhomogeneity parameters.

The obtained modeling confirms the validity and importance of introducing new parameters of textile structure inhomogeneity, both in terms of IAR and IER variability assessment. The model identified the most important dependence of the IAR and IER inhomogeneity equation for (${\bar{V}}_{IAR1}$) (15) and for (${\bar{V}}_{IER2}$) (20) as the most significant method for calculating the main indicator for individual parameters of inhomogeneity of the size, location, and shape of fabric structural elements among other calculation methods. Model 2, similar to Model 1, also selected the most important dependencies of the S equation for the ITP area shape (2) from among the proposed equations S and S1, which are presented in Equations (2) and (3).

In order to verify the new method, multiple regression was also performed using previously employed methods to assess the uniformity of the fabric structure, such as pore size distribution and variability of the warp and weft pitches.

### 3.6. Verification of the New Method Using the Multiple Regression Models of the Air Flow AirF [mm/s] and the Pore Size Distribution Parameters

The multiple regression models of airflow AirF [mm/s] and the statistical description of the pore size distribution were examined. This multiple regression was performed using global parameters of pore size distribution as previous indicators used for uniformity of structure to verify the new methodology of IAR and IER inhomogeneity parameters. The multiple regression analysis aimed to explore the relationship between airflow AirF [mm/s] as the output variable and the statistical description of the pore size distribution, including (min (A), max (A), average (A), median \A\, and standard deviation σ (A)) as input variables. The input variables consisted of a set of five variables, including their first to third power forms and their 18 mutual interactions. AirF = 201.36 + (0.10 × St Dev σ (A))(Model 5)


In Model 5, (R^2^) = 0.72 and adjusted (R^2) = 0.71 with F = 10. The strongest connections between individual independent variables and the dependent variable AirF are observed for the standard deviation of the average pore size (A) of the standard deviation of the size pore distribution (St Dev σ (A)) (BETA = 0.84), *p* = 0.0000.

Then, a multiple regression of the (St Dev σ (A)) was performed for the plain fabric structure ${\bar{V}}_{IAR}$ and ${\bar{V}}_{IER}$ homogeneity parameters.

### 3.7. Verification of the New Method Using the Multiple Regression Models of the St Dev σ (A) and ${\bar{V}}_{IAR}$ and ${\bar{V}}_{IER}$ Inhomogeneity Fabric Structure Parameters

This multiple regression was performed for global parameters of size pore distribution used as an indicator of uniformity of structure in order to verify the new methodology of IAR and IER inhomogeneity parameters. The multiple regression analysis aimed to explore the relationship between standard deviation of the pore size distribution St Dev σ (A) as the output and various parameters of plain fabric structure inhomogeneity, including both ${\bar{V}}_{IAR}$ and ${\bar{V}}_{IER}$ (V_IAR__A, V_IAR__Feret, V_IAR__AspR, V_IAR__FormF, V_IAR__P_wa_, V_IAR__P_we_, V_IAR__D_,_ V_IAR__RID, V_IAR__S, V_IER__A, V_IER__Feret, V_IER__AspR, V_IER__FormF, V_IER__P_wa_, V_IER__P_we_, V_IER__D_,_ V_IER__RID, and V_IER__S) as input variables. The input variables consisted of a set of 54 variables, including their first to third power forms and their 108 mutual interactions.
St Dev σ (A) = 1178.26 + (112.10 × V_IAR__P_wa_)(Model 6)

In Model 6, (R^2^) = 0.86, (R^2) = 0.85, and F = 10 were obtained. The strongest connections between individual independent variables and the dependent variable St Dev σ (A) are observed in these cases for V_IAR__P_wa_ (BETA = 0.927), *p* = 0.0000.

These results are the same as those obtained in Model 3 for the dependent variable AirF, which confirms the correctness of the verification.

### 3.8. Verification of the New Method Using the Multiple Regression Models for the Air Flow AirF [mm/s] and the Warp and Weft Pitches

This multiple regression was performed for global parameters of the weft and warp pitches used as an indicator of uniformity of structure in order to verify the new methodology of inter- and intra-repeat homogeneity parameters. The multiple regression analysis aimed to explore the relationship between air flow AirF [mm/s] as the output variable and statistical description of the pore size distribution, including average (P_wa_), (P_we_), and standard deviation (P_wa_), (P_we_) as input variables. The input variables consisted of a set of 4 variables, including their first to third power forms and their 12 mutual interactions.AirF = 255.33 + (10.97 × St Dev (P_wa_))(Model 7)

In Model 7, (R^2^) = 0.67 and adjusted (R^2) = 0.66 with F = 10. The strongest connections between individual independent variables and the dependent variable AirF are observed for the standard deviation of the average warp pitches (St Dev (P_wa_)) (BETA = 0.82). *p* = 0.0000.

Then, a multiple regression of the (St Dev (P_wa_)) was performed for the ${\bar{V}}_{IAR}$ and ${\bar{V}}_{IER}$ inhomogeneity fabric structure parameters.

### 3.9. Verification of the New Method Using the Multiple Regression Models for the St Dev (P_wa_) and ${\bar{V}}_{IAR}$ and ${\bar{V}}_{IER}$ Inhomogeneity Fabric Structure Parameters

This multiple regression was performed for global parameters of the warp pitches’ standard deviation used as an indicator of uniformity of structure to verify the new methodology of IAR and IER inhomogeneity parameters. The multiple regression analysis aimed to explore the relationship between the standard deviation of the pore size distribution St Dev (P_wa_) as the output and various parameters of the plain fabric structure homogeneity, including both ${\bar{V}}_{IAR}$ and ${\bar{V}}_{IER}$ (V_IAR__A, V_IAR__Feret, V_IAR__AspR, V_IAR__FormF, V_IAR__P_wa_, V_IAR__P_we_, V_IAR__D_,_ V_IAR__RID, V_IAR__S, V_IER__A, V_IER__Feret, V_IER__AspR, V_IER__FormF, V_IER__P_wa_, V_IER__P_we_, V_IER__D_,_ V_IER__RID, and V_IER__S) as input variables. The input variables consisted of a set of 54 variables, including their first to third power forms and their 108 mutual interactions.
St Dev (P_wa_) = 4.23 + (1.00 × V_IAR__P_wa_) + (V_IER__ P_we_^2^ × V_IER__D_ITP_)(Model 8)

In Model 8, (R^2^) = 0.98 and adjusted (R^2) = 0.97 with F = 10. The strongest connections between individual independent variables and the dependent variable St Dev (P_wa_) are observed in these cases for the IAR inhomogeneity of the warp pitches V_IAR__P_wa_ (BETA = 0.961), p = 0.0000, IER inhomogeneity of the warp pitches, and the ITP position (V_IER__P_we_^2^ × V_IER__D_ITP_) (BETA = 0.098), *p* = 0.0037.

### 3.10. Verification of the New Method Using the Multiple Regression Models for the Air Flow AirF [mm/s] and and the General Coefficients of IAR and IER Inhomogeneity (${\bar{V}}_{IAR}{,\bar{V}}_{IER}$) and Statistical Description of the Pore Size Distribution and Warp and Weft Pitches

This multiple regression was performed for the verification of the general coefficients of IAR and IER inhomogeneity (V¯IAR,V¯IER) together with parameters of pore size distribution and the warp and weft pitches. The multiple regression analysis aimed to predict the air flow AirF [mm/s] (output variable) based on the general coefficients of IAR and IER inhomogeneity V¯IAR,V¯IER and the statistical description of the pore size distribution and the warp and weft pitches. These parameters include V¯IAR115;V¯IAR216;V¯IAR317;V¯IER119;V¯IER220;V¯IER3(21); (min (A); max (A); average μ (A); median \A\; standard deviation σ (A)); and average and standard deviation (P_wa_), (P_we_) (input variables). The input variables comprised 40 parameters in the form of the first to the third power, along with their 48 mutual interactions. The strongest dependencies for F = 10 are presented below.
(Model 9)AirF=105.53+(13.34×V¯IAR1)−(3.98×V¯IER2)

In Model 9, R^2^ = 0.81 and adjusted R^2 = 0.79 with F = 10. The most significant correlations with the dependent variable AirF are (${\bar{V}}_{IAR1}$) (BETA = 1.00), p = 0.0000 and (${\bar{V}}_{IER2}$) (BETA = − 0.31), p = 0.0031.

In the final model, Model 9, as in Model 4, the most significant correlations were found with the general coefficients of IAR and IER inhomogeneity, (${\bar{V}}_{IAR1})$ and (${\bar{V}}_{IER2}),$ among all the input variables used in this correlation, including both new and earlier methods. These coefficients have the most significant impact on the air flow parameter. The new general coefficients of IAR and IER inhomogeneity, (${\bar{V}}_{IAR1})$ and (${\bar{V}}_{IER2}),$ cover a wider range of structural parameters, including the size, shape, and position of the ITPs as well as the warp and weft pitches. Additionally, they allow for the assessment of inhomogeneity in terms of both IAR and IER.

The performed regression models highlight the importance and correlation of the weave parameters IAR and IER with the actual airflow parameter, as well as with the standard deviation of the pore size distribution and the matrix pitch. Mutual correlations and key dependencies of these parameters were determined. The models identified the most important structural parameters, including size (V_IAR__A), shape (V_IER__AspR), and position of ITPs (V_IER__D_ITP_), as well as warp (V_IAR__P_wa_) and weft (V_IER__P_we_) pitches, in the context of both IAR and IER inhomogeneity.

Most importantly, in the final model, the correlation with AirF was determined from the entire pool of all parameters of the new and earlier methods, in which the new general coefficients of IAR and IER inhomogeneity, (${\bar{V}}_{IAR1}$) and (${\bar{V}}_{IER2}$), turned out to be the most important. These coefficients cover a wider range of structural parameters, including the size, shape, and position of the ITPs as well as the warp and weft pitches. Additionally, they allow for the assessment of inhomogeneity in terms of both IAR and IER.

It should be noted that the pore size distribution method provides information on the overall distribution of irregularities for only one structural parameter, i.e., pore size, and only in terms of global distribution, without specifying the location of these irregularities in the fabric structure. This method is useful at the initial stage of identifying differences in structures. Similarly, the warp and weft pitches’ variability parameter, without specifying the location, provides preliminary information about the general nature of the inhomogeneity.

The new IAR and IER inhomogeneity methodology allows for a comprehensive assessment of the fabric structure, considering the size, shape, and position of the ITPs as well as the warp and weft pitches. It also enables the creation of an average bimodal grid and model of the ITPs using these real structural parameters during processing and image analysis of the fabric structure.

Multiple regression models of the airflow (AirF [mm/s]) with inhomogeneity parameters highlight the importance of factors such as the size, shape, and location of the ITPs as well as the size and location of the weft and warp pitches. Location parameters play a key role in this methodology, enabling each structural parameter to be associated with a specific module within a weave fabric repeat.

## 4. Conclusions

A novel methodology for objectively measuring fabric structure homogeneity has been developed, introducing two new general coefficients: IAR inhomogeneity (${\bar{V}}_{IAR}$) and IER inhomogeneity (${\bar{V}}_{IER}$). These coefficients are derived from structural homogeneity parameters such as the size, shape, and position of individual ITPs as well as the values and positions of the warp and weft pitches assigned to each repeat in the image. They provide a comprehensive assessment of fabric structure variability and irregularity within and between repeats, offering specific metrics for quantifying homogeneity across repeats.

A new method for assessing the IAR and IER inhomogeneity of plain fabric structure was verified using multiple regression. Airflow multiple regression models (AirF [mm/s]) highlight the importance of the size, shape, and location of the ITPs as well as the size and location of the weft and warp pitches. Location parameters play a key role in this methodology, enabling each structural parameter to be associated with every fabric repeat in the fabric structure.

Existing methods, such as pore sizer distribution or variability of the warp or weft pitches, provide an initial, general diagnosis of a given structure, but without precise identification of where a given inhomogeneity occurs and what causes it. In the first case, these methods concern only the size of the ITP without information about its shape and location. However, in the second case, we only have information about the size of the pitches, also without information about their location. However, the new methodology allows one to precisely identify the variability of structural parameters within each subsequent weave fabric repeat and the variability between these repeats. The structural parameters of ITPs are described by size, shape, and location and are precisely embedded in the grid of localized warp and weft pitches.

The importance of the developed methodology is particularly important for the precise modeling of filter structures, barriers, and frames of composite products. These products depend on a homogeneity structure to maintain filtration quality and mechanical properties during composite formation. Even minor structural variations, either within or between weave fabric repeats, can cause fluctuations in filtration efficiency and mechanical properties, especially in composite products where fabric structures form the core.

These general coefficients complement existing structure parameters and offer a practical framework for evaluating fabrics used in specialized applications like filtration, barrier, and composite fabrics. The future direction of this method lies in its automation, facilitating its direct implementation during fabric production.

In summary, accurate fabric structural analysis, focusing on the location, size, and shape of ITPs, as well as the pitch values and locations of both thread systems, is essential. The newly introduced inhomogeneity parameters, validated against airflow parameters, underscore the significance of analyzing IAR and IER variability. This method serves as a vital tool for detailed fabric structural analysis, demonstrating its effectiveness across a diverse range of 30 plain weave fabrics and likely applicable to other weave types as well.

## 5. Patents

The method has been patented in the Patent Office RP.

## Figures and Tables

**Figure 1 materials-17-03229-f001:**
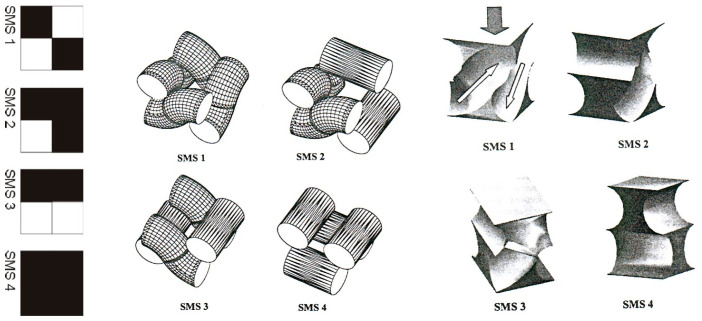
The weaving notation of weave structural modules (SMS) and their inter-thread spaces in the fabric [14].

**Figure 2 materials-17-03229-f002:**
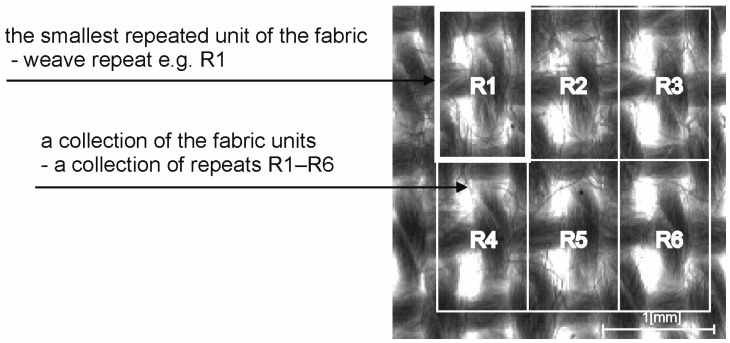
Graphical visualization of the fabric weave repeat (e.g., R1) and collection of the repeats (R1–R6) separated in the image.

**Figure 3 materials-17-03229-f003:**
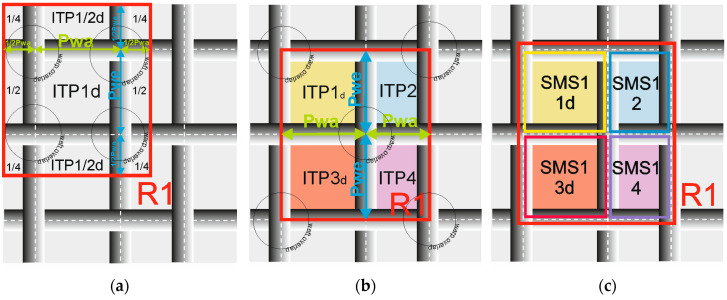
Plain weave repeat: (**a**) classic repeat R1, (**b**) repeat R1 for homogeneity analysis, (**c**) weave structural modulus SMS1 type in the repeat R1 for analysis, where R1 (red square)—a repeat of weave; P_wa_, P_we_ (green and blue arrow)—warp, weft pitches; ITP1d, ITP2, ITP3d, ITP4 (different colored squares)—ITPs in the repeats; SMS1_d, SMS1_2, SMS1_3d, SMS1_4—SMS1 modulus types in the repeat R1; and grey lines with circles—warp, weft threads with warp and weft overlaps.

**Figure 4 materials-17-03229-f004:**
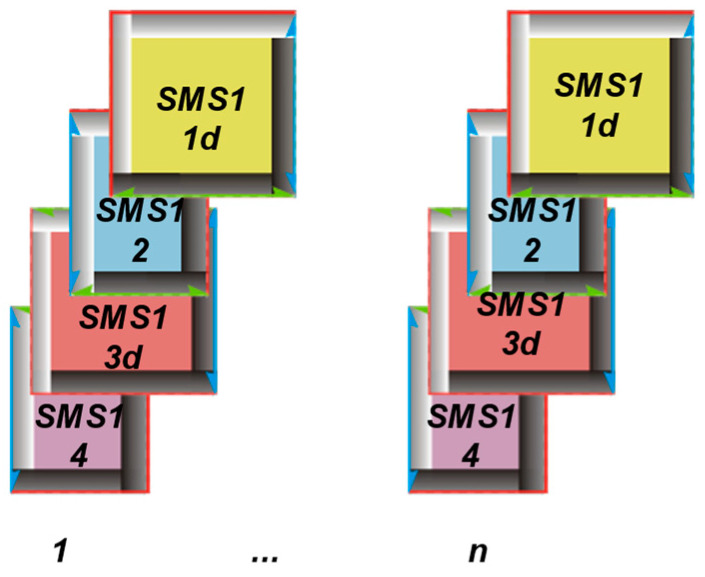
The graphical visualization for calculating the IAR inhomogeneity, where n represents the number of analyzed repeats with the structural modules SMS1{1d, 2, 3d, 4}.

**Figure 5 materials-17-03229-f005:**
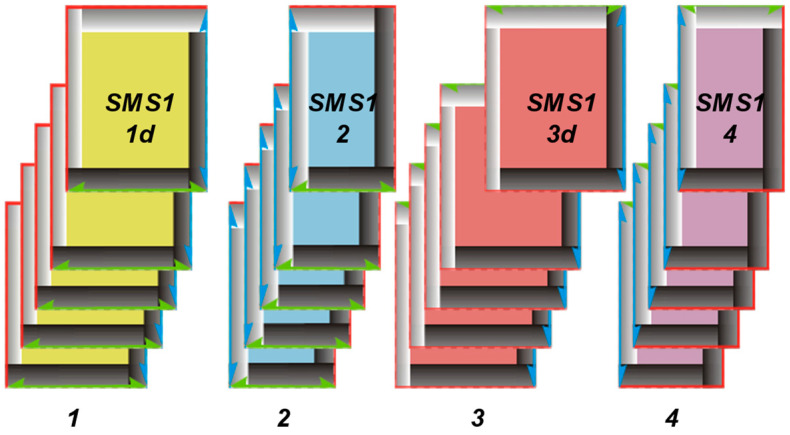
The graphical visualization for calculating the inter-repeat inhomogeneity (IER), where 4 represents the number of structural modules SMS1{1d, 2, 3d, 4} in the plain weave fabric repeat.In this research, ITP refers to the inter-thread porosity, encompassing the hairiness of the thread. ITP represents the spaces or gaps between adjacent threads, such as warp and weft threads, in the woven structure. These pores are formed during the weaving process as threads interlace to create the fabric. The size, shape, and location of ITPs can vary based on factors like the weave pattern, thread density, and weaving parameters such as tension.

**Figure 6 materials-17-03229-f006:**
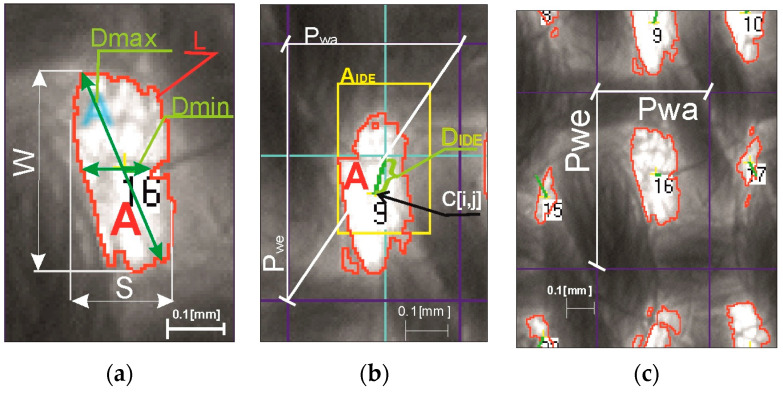
Illustration of the parameters essential for fabric structure homogeneity: (**a**) ITP size: A: area, W: height, S: width; ITP shape: Dmax: maximum area diameter, Dmin: minimum area diameter, L: area perimeter; (**b**) ITP location: 16, 9: consecutive ITP numbers, A_IDE_: ideal area (in the average grid), D_IDE_: diameter of ideal area (in the average grid), C[i,j]: centre of gravity of the ITP; (**c**) the thread pitches: P_wa_, P_we_: warp and weft pitches [17].

**Figure 7 materials-17-03229-f007:**
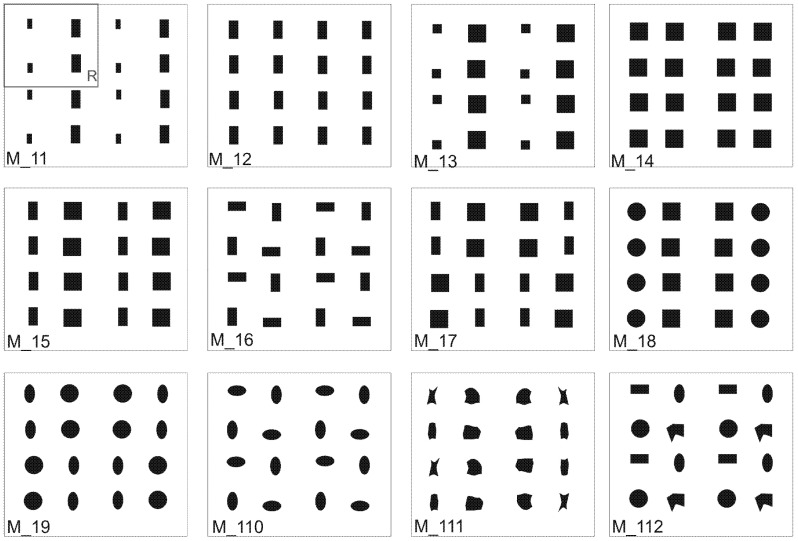
Images of theoretical fabric models differentiated by the size, shape, and location of artificial ITPs.

**Figure 8 materials-17-03229-f008:**
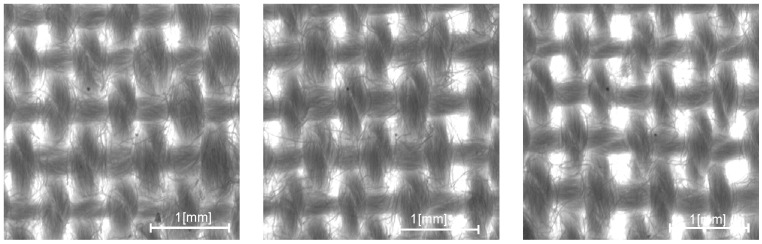
Images of three characteristic structures of plain cotton fabrics with different groupings of warp threads, representing a group of 30 fabrics produced on a laboratory loom by varying loom settings.

**Figure 9 materials-17-03229-f009:**
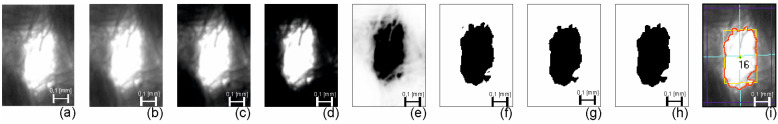
The image analysis of the optimal algorithm of tested fabrics with plain weaves focusing on the ITP: (**a**) the picture after the acquisition, (**b**) low-pass filtering, (**c**) histogram equalization, (**d**) non-linear filtration (x^2^), (**e**) image negative, (**f**) thresholding operation with the auto threshold set by copyright procedure, (**g**) closing operation, (**h**) opening operation, and (**i**) cluster analysis automatically set by copyright procedure for segmentation recognition and the classification and interpretation of ITPs [18].

**Figure 10 materials-17-03229-f010:**
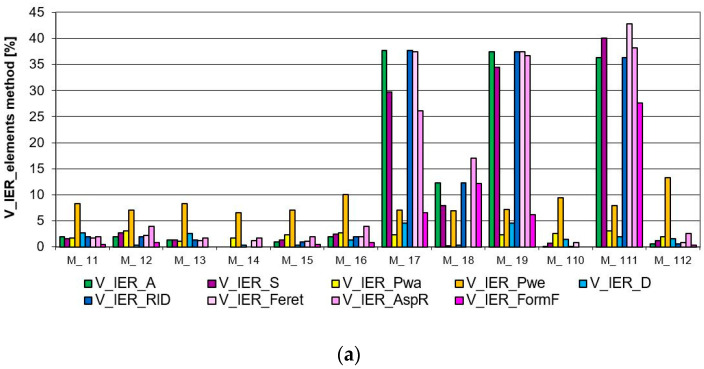
Results of the analysis of artificial image models for verification of IER inhomogeneity parameters: (**a**) “elements” method, and (**b**) “average” method.

**Figure 11 materials-17-03229-f011:**
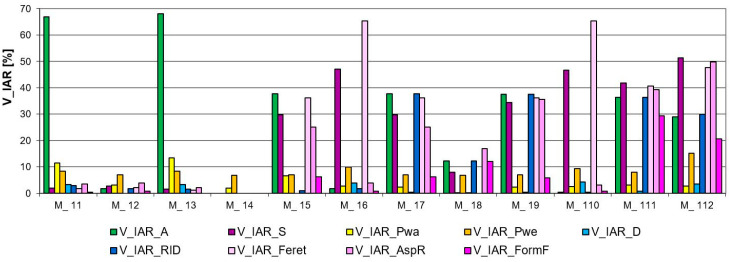
The results of the analysis of artificial image models for verification of IAR inhomogeneity parameters.

**Figure 12 materials-17-03229-f012:**
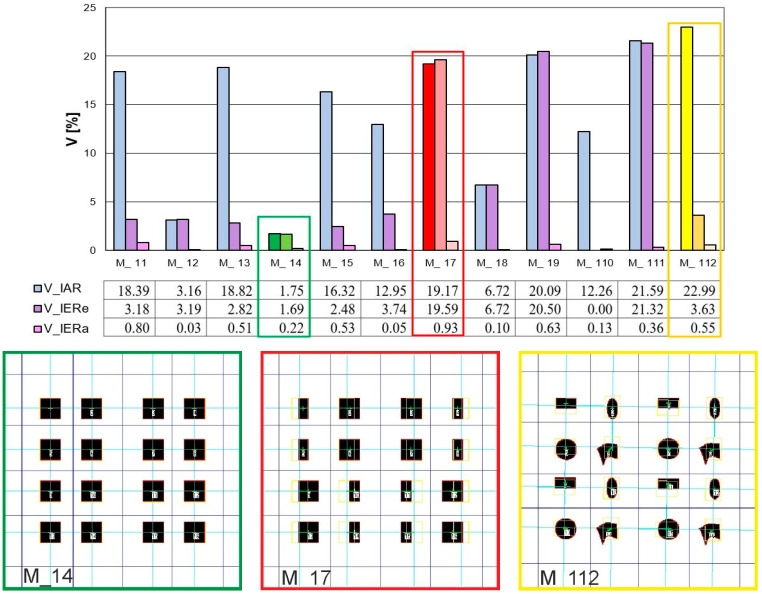
Results of verification of the IAR (${\bar{V}}_{IAR}$) and IER (${\bar{V}}_{IER}$) inhomogeneity in artificial images, where: green color—M_14 with the best homogeneity results (low level of ${\bar{V}}_{IAR}=1.75\%$ and ${\bar{V}}_{IER}=1.69\%$); red color—M_17 with the worst homogeneity result (high level of ${\bar{V}}_{IAR}=19.17\%$ and ${\bar{V}}_{IER}=19.59\%$); yelow color—M_112 with disturbed homogeneity (high level of ${\bar{V}}_{IAR}=22.99\%$ and low level ${\bar{V}}_{IER}=3.63\%$).

**Figure 13 materials-17-03229-f013:**
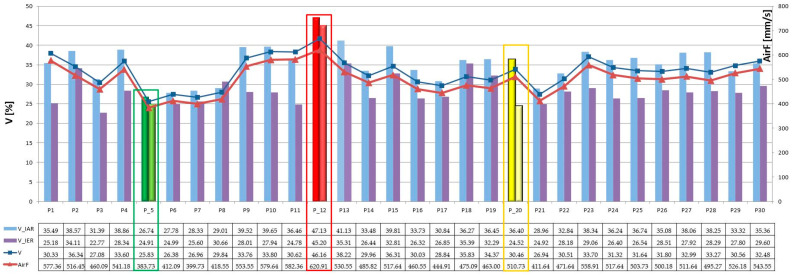
Results of IAR $\left({\bar{V}}_{IAR}\right)$ and IER (${\bar{V}}_{IER})$ inhomogeneity on the graph for all fabrics, and results of verification of (AirF) [mm/s] airflow for all plain fabrics, where: green color—P_5 fabric with the best homogeneity results (low level of ${\bar{V}}_{IAR}=26.74\%$ and ${\bar{V}}_{IER}=24.91\%$, AirF = 383.73 mm/s); red color—P_12 fabric with the worst homogeneity result (high level of ${\bar{V}}_{IAR}=47.13\%$ and ${\bar{V}}_{IER}=45.20\%,AirF=620.91\;mm/s$); yelow color—P_20 fabric with disturbed homogeneity (high level of ${\bar{V}}_{IAR}=36.40\%$ and low level ${\bar{V}}_{IER}=24.52\%,$ AirF = 510.73 mm/s).

**Figure 14 materials-17-03229-f014:**
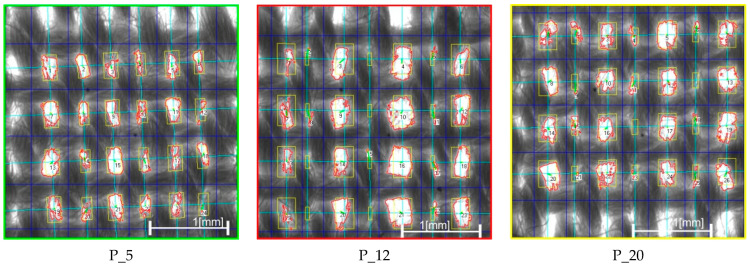
Images of three characteristic plain fabrics after image analysis: P_5 with the best homogeneity (green color), P_12 with the worst homogeneity (red color), and P_20 with the best IER homogeneity but the worst IAR homogeneity (yelow color).

**Figure 15 materials-17-03229-f015:**
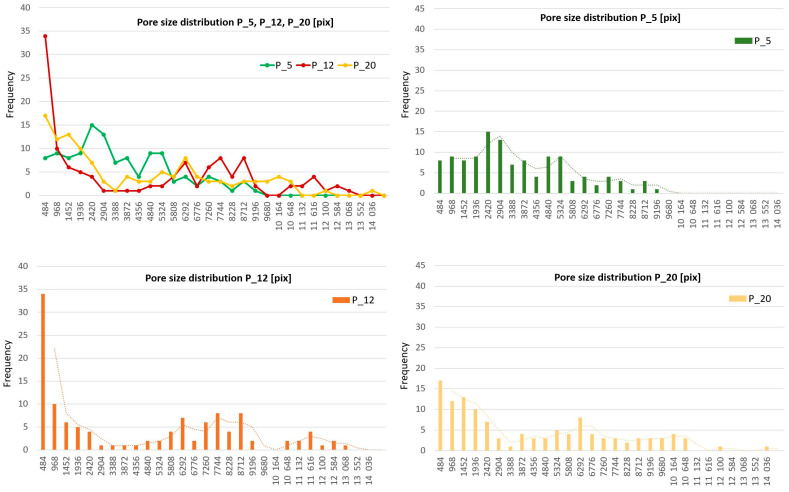
After image analysis, the pore size distribution for three characteristic plain fabrics: P_5, P_12, P_20.

**Table 1 materials-17-03229-t001:** Measured parameters of theoretical fabric models: A–average of area ITPs, P_wa_, and P_we_ [pix]–average of warp and weft pitches and their standard deviations δ (A), δ (P_wa_), δ (P_we_) [pix].

	M_11	M_12	M_13	M_14	M_15	M_16	M_17	M_18	M_19	M_110	M_111	M_112
A	1159	1830	2265	3600	2715	1830	2715	3254	2196	1481	1698	2121
δ (A)	671	30	1335	0	885	30	885	346	712	5	536	532
P_wa_	147	148	146	144	146	147	146	144	146	146	148	145
δ (P_wa_)	15	4	17	3	9	4	3	0	3	3	4	4
P_we_	138	138	138	138	138	139	138	138	138	138	138	138
δ (P_we_)	10	9	10	9	9	12	9	8	9	11	10	24

**Table 2 materials-17-03229-t002:** Measured properties of yarns.

Property of Yarns	Value
Cv_m_ Variation of linear mass	13.93%
Warp twist	646 S twist/m
Weft twist	604 S twist/m
Hairiness	7.31 fibers/m
Number of thin places per 1000 m	<5%
Number of thick places per 1000 m	<25%

**Table 3 materials-17-03229-t003:** Measured parameters of plain fabrics: A–average of area ITPs, P_wa_, and P_we_ [pix]–average of warp and weft pitches and their standard deviations δ (A), δ (P_wa_), δ (P_we_) [pix].

**Plain Fabrics**	**P1**	**P2**	**P3**	**P4**	**P5**	**P6**	**P7**	**P8**	**P9**	**P10**	**P11**	**P12**	**P13**	**P14**	**P15**
A	3554	3508	3918	3816	3338	3384	3437	3161	3939	3785	3863	4029	3527	3741	3155
δ (A)	2991	3590	2808	3593	2174	2257	2139	2321	3650	3608	3196	3848	3648	3027	3237
P_wa_	134	137	133	133	134	134	135	133	134	133	134	135	134	133	134
δ (P_wa_)	21	26	18	25	13	15	15	16	26	28	24	29	30	18	26
P_we_	218	226	220	225	216	221	216	223	214	213	213	217	218	217	216
δ (P_we_)	12	9	11	15	8	8	13	12	11	8	14	12	13	11	9
**Plain Fabrics**	**P16**	**P17**	**P18**	**P19**	**P20**	**P21**	**P22**	**P23**	**P24**	**P25**	**P26**	**P27**	**P28**	**P29**	**P30**
A	3170	3342	2726	3282	3903	3579	3652	3674	3709	3871	3817	4135	4165	3830	3708
δ (A)	2622	2430	2526	2964	3359	2409	2759	3426	2976	3434	3186	3669	3595	3095	3408
P_wa_	135	137	133	133	132	135	134	135	133	135	136	137	133	133	135
δ (P_wa_)	21	20	23	23	24	18	17	27	23	24	22	27	25	20	24
P_we_	214	214	212	214	211	216	227	213	214	214	215	216	217	220	217
δ (P_we_)	10	11	12	9	9	11	12	11	14	11	10	15	11	8	13

**Table 4 materials-17-03229-t004:** The statistical description of the pore size distribution for every plain fabric: min (A), max (A), average (A), median \A\, and standard deviation σ (A), where A–area ITPs [pix].

	**P_1**	**P_2**	**P_3**	**P_4**	**P_5**	**P_6**	**P_7**	**P_8**	**P_9**	**P_10**	**P_11**	**P_12**	**P_13**	**P_14**	**P_15**
min (A)	68	1	182	64	36	288	216	176	82	50	58	1	1	80	64
max (A)	12,144	13,906	10,220	13,090	8822	8710	8220	9818	12,706	13,654	10,938	12,716	12,608	10,564	14,520
(A)	3554	3508	3918	3816	3338	3384	3437	3161	3939	3785	3863	4029	3527	3741	3155
\A\	2711	1862	3220	2340	2862	3161	2994	2338	2641	2597	2487	2838	2097	3073	1516
σ (A)	2983	3566	2785	3582	2168	2240	2136	2317	3642	3600	3199	3853	3627	3004	3312
	**P_16**	**P_17**	**P_18**	**P_19**	**P_20**	**P_21**	**P_22**	**P_23**	**P_24**	**P_25**	**P_26**	**P_27**	**P_28**	**P_29**	**P_30**
min (A)	154	1	58	1	92	46	84	94	70	92	110	104	1	78	114
max (A)	12,938	12,116	10,342	10,562	13,698	10,512	11,828	12,310	9950	11,540	11,972	13,240	12,870	12,242	11,488
(A)	3170	3342	2726	3282	3903	3579	3652	3674	3709	3871	3817	4135	4165	3830	3708
\A\	2358	2812	2018	2274	2481	3256	3239	2259	3129	2559	2804	2838	2898	3077	2538
σ (A)	2605	2430	2518	2953	3357	2407	2792	3420	2976	3461	3169	3643	3589	3112	3391

## Data Availability

The data presented in this study are available upon request from the corresponding author.

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
