# Peer review of "A New Method for Evaluating the Homogeneity within and between Weave Repeats in Plain Fabric Structures Using Computer Image Analysis"

_materials, 2024, doi:10.3390/ma17133229_

Round 1

Reviewer 1 Report

Comments and Suggestions for Authors

The paper lacks a comparative analysis of the detection results and pore size distribution. It is recommended to use equipment such as PMI to detect the actual pore size distribution of fabrics to verify the effectiveness of the method.

The paper uses air permeability as a comparison, and the detection range of air permeability is inconsistent with the image acquisition range of 3 × 3m2 in this paper. How to explain the impact of the detection range on the results?

Suggest that the title of the paper reflect plain weave fabrics, as the samples used in the paper are all plain weave fabrics and cannot verify their effectiveness on other structural fabrics. This method is difficult to accurately detect fabrics with holes that cannot be realistically detected in images.

Comments on the Quality of English Language

English writing can be further optimized.

Author Response

Thank you very much for your questions and correction. It is very important for me. Below are the comments and replies to them.

The paper lacks a comparative analysis of the detection results and pore size distribution. It is recommended to use equipment such as PMI to detect the actual pore size distribution of fabrics to verify the effectiveness of the method.

An image analysis was performed to analyze size pore distribution because we cannot access equipment such as PMI. The description and results are inserted under the results of the new homogeneity method in Figures 15-16. In chapter 3.2.

The paper uses air permeability as a comparison, and the detection range of air permeability is inconsistent with the image acquisition range of 3 × 3m2 in this paper. How to explain the impact of the detection range on the results?

The author’s program, MagFABRIC, allows one to stitch several images together to enlarge the fabric surface while maintaining the highest possible accuracy at a magnification of 3x3 mm². Image acquisition was performed in this way: 30 photos were taken along a diagonal line across the entire width of the 1.5 m long fabric. However, at each acquisition site, several adjacent images were stitched together to obtain a larger research area. The FX 3300 III air permeability tester was used with a 20 mm² round test head.

Added Information about the FX 3300 III Test Area and Image Stitching:

In Chapter 2.2.2:

The author’s program, MagFABRIC, allows one to stitch several images together to enlarge the fabric surface while maintaining the highest possible accuracy at a magnification of 3x3 mm².

In Chapter 3.2:

The research area was determined by the 20 mm² circular testing head.

Suggest that the title of the paper reflect plain weave fabrics, as the samples used in the paper are all plain weave fabrics and cannot verify their effectiveness on other structural fabrics. This method is difficult to accurately detect fabrics with holes that cannot be realistically detected in images.

The term plain weave has been added to the article title.

English writing can be further optimized.

The article has been revised for better English.

Reviewer 2 Report

Comments and Suggestions for Authors

Previous research confirms the necessity of conducting a thorough and precise analysis of fabric structure homogeneity." Which researches?

Each model in Figure 7 must be specified in the legend.

1.25x likely refers only to the objective lens magnification. What is the actual magnification of the image?

Based on the analyses presented, the method shows repeatability; however, it must be validated through comparisons with other established methods. The analyses need to be contrasted and validated according to their correlation.

Author Response

Thank you very much for your questions and correction. It is very important for me. Below are the comments and answers to them.

Previous research confirms the necessity of conducting a thorough and precise analysis of fabric structure homogeneity." Which researches?

This sentence was corrected.

Each model in Figure 7 must be specified in the legend.

This description was added in Table 1.

1.25x likely refers only to the objective lens magnification. What is the actual magnification of the image?

Yes, the 1.25x magnification belongs to the lens. The image was transmitted directly by the camera, not through an eyepiece that could increase magnification. However, the camera did not zoom in. The most important goal in image acquisition was to include the report in the image along with visible highlights. Tests were carried out in which a magnification of 1.25 and an image range of 3x3mm gave the best results in measuring uniformity.

Based on the analyses presented, the method shows repeatability; however, it must be validated through comparisons with other established methods. The analyses need to be contrasted and validated according to their correlation.

Verification of the new method was extended in the chapters 3.3 – 3.10  using the multiple regression models for the airflow AirF[mm/s] and statistical description of the pore size distribution and warp and weft pitches.

Reviewer 3 Report

Comments and Suggestions for Authors

The topic is interesting and related to the scope of this journal.

The following corrections need to be done:

1. The section "2.2.2. The Images of Plain Weave Fabrics for Verification of the New Method" needs to be rewritten. The following segments need to be corrected:

- "produced on a laboratory loom" - the name/producer of loom should be given

- "using combed cotton and double ring thread" - the correct technical terms need to be used. Instead of thread, the correct word is yarn. Instead of "double ring" should be the correct term; in this case I believe it refers to two-ply ring yarn.

2. Table 1 should be significantly corrected.

- No need to give warp and weft densities, as data is given in the text. Linear mass and number of twist is also given in the text, so it is important to avoid repetitions.

- As per yarn evenness results, is should be "Number of thin places per 1000 m" instead of Thin per 1000 m.

- For the hairiness add the word index if this is applicable for the result given. - Acquisition parameters should be given in text only, not in the table.

- It stays unclear which are exact differences between 30 produced fabrics. The data on these 30 fabrics should be given within the Table 1, so readers can get info on fabric properties.

3. Figure 8 should be excluded. It is more important to give details on produced fabrics (in Table 1) than to present the images that are rather similar.

4. Check the Figure in Line 521. Where is the title?

5. The conclusion should be focused at the exact description of the improvements in methodology that is given within this paper (compared to previously published methods).

6. The literature overview is rather limited. Many references are outdated. Newer references and larger number of references should be included in the paper and the Introduction should be rewritten accordingly. The references are not written according to the instructions.

Author Response

Thank you very much for your questions and correction. It is very important for me. Below are the comments and answers to them.

The topic is interesting and related to the scope of this journal.

The following corrections need to be done:

  1. The section "2.2.2. The Images of Plain Weave Fabrics for Verification of the New Method" needs to be rewritten. The following segments need to be corrected:

- "produced on a laboratory loom" - the name/producer of loom should be given

This description was corrected and the name and producer of the loom were added: Saurer 100W Shuttle Loom by Saurer Group.

- "using combed cotton and double ring thread" - the correct technical terms need to be used. Instead of thread, the correct word is yarn. Instead of "double ring" should be the correct term; in this case I believe it refers to two-ply ring yarn.

This description was corrected.

  1. Table 1 should be significantly corrected.

- No need to give warp and weft densities, as data is given in the text. Linear mass and number of twist is also given in the text, so it is important to avoid repetitions.

This description in Table 2 was corrected.

- As per yarn evenness results, is should be "Number of thin places per 1000 m" instead of Thin per 1000 m.

This description in Table 2 was corrected.

- For the hairiness add the word index if this is applicable for the result given. - Acquisition parameters should be given in text only, not in the table.

This description in Table 2 was corrected.

- It stays unclear which are exact differences between 30 produced fabrics. The data on these 30 fabrics should be given within the Table 1, so readers can get info on fabric properties.\

This description about difference of these fabric has added below Table 2 and added the global parameters of these fabrics Table 3.

  1. Figure 8 should be excluded. It is more important to give details on produced fabrics (in Table 1) than to present the images that are rather similar.

In the Figure 8 was changed the 30 images on 3 characteristic images and added the Table 3 with description of these fabrics.

  1. Check the Figure in Line 521. Where is the title?

The title was added for the Figure 13 and 14 separately.

  1. The conclusion should be focused at the exact description of the improvements in methodology that is given within this paper (compared to previously published methods).

The conclusion was corrected.

  1. The literature overview is rather limited. Many references are outdated. Newer references and larger number of references should be included in the paper and the Introduction should be rewritten accordingly. The references are not written according to the instructions.

The literature was added. The references was corrected by the style Earthquake Engineering & Structural Dynamics according to Mendeley Cite.

Round 2

Reviewer 1 Report

Comments and Suggestions for Authors

Accept.

Comments on the Quality of English Language

Accept.

Author Response

Thank you very much for your comments on English. I once again analyzed the text in this respect and gave it to a person fluent in English to review. I found the following errors which I corrected. I hope I managed to spot all the shortcomings.

36- in their in

58- The authors of the article [8], took

88-89 was … calculated average inter-yarn pore area and derived intra-yarn porosity derived from equations.

95 ….whereas the threshold in the algorithm had a minor influence

99 However in the article [13]  there was presented  features a research

105 The article by [14] underscored

117-118 The article by  [15]) investigated the thread channels in woven fabric structures, building upon the modules identified by in [14].

119 Key parameters determined  included channel height

126 Preliminary research in this domain is outlined in the article by [16],

128 The evaluation was centered

272 for the overall assessment of of IAR inhomogeneity ).

386 and the large importance the large described in the earlier author's article [18].

408 The uniformity  criterion was criteria ware the segmentation threshold and

419 Cluster analysis with the author’s modernization provides an optimal approach

434 Noteworthy is Model 112, is noteworthy because it which has the highest (VIER_Pweft) index of 13 %, which is caused by the presence of various shapes of elements with different locations of centers of gravity in the image.

466 The next two high peaks in the graph are due to the elongation coefficient

477 formulas (4.2 and 4.7 respectively),  respectively, effectively reflects

478 Consistently with the

514 and strong stabiliszation occurring in the area of weaving.

820 Airflow Multiple regression models of the Airflow (AirF [mm/s])

838 However, the new methodology allows to for precisely